# Healthcare worker's emotions, perceived stressors and coping mechanisms during the COVID-19 pandemic

**Suzanne Rose[1]\*, Josette Hartnett[1], Seema Pillai[2]**

**1** Office of Research and Clinical Trials, Stamford Hospital, Stamford, Connecticut, United States of America,
**2** Department of Nursing, Stamford Hospital, Stamford, Connecticut, United States of America

\* srose@stamhealth.org

**Data Availability Statement:** All files are available from a data repository under the DOI (DOI: doi.org/10.3886/E142222V1).

**Funding:** The authors received no specific funding for this work.

## Abstract

Increasing cases, insufficient amount of personal protection equipment, extremely demanding workloads, and lack of adequate therapies to save lives can contribute to a psychological burden directly related to working during disease outbreaks. Healthcare workers (HCWs) are at a high risk of contracting COVID-19 due to its ability to spread efficiently through asymptomatic and symptomatic individuals. There are limited studies assessing the pandemic's psychological impact on HCWs, specifically those in close proximity to hospitalized patients with COVID-19. Our study explored the emotions, perceived stressors, and coping strategies of front-line HCWs at high risk of exposure to COVID-19 during the first surge at our community-based teaching hospital, the epicenter of COVID-19 in Connecticut. A validated comprehensive questionnaire derived and modified from previous epidemics was used to inquire about staff feelings, factors that caused stress and factors that mitigated stress. Personal coping strategies and factors that can increase staff's motivation to work during future events of similar nature were also asked. Emotional reactions, coping mechanisms, and stressors varied by healthcare role, while some experiences and reactions were similar among groups. Willingness to participate in a second wave of the pandemic or future outbreaks is strongly driven by adequate personal protective equipment, financial recognition, and recognition from management, similarly reported in previous disease outbreaks. All groups felt a reduction in stress due to a sense of camaraderie and teamwork, as well as when sharing jokes or humor with colleagues. Our HCWs at high risk of exposure experienced significant emotional distress during the first wave of the COVID-19 pandemic. By understanding the needs and experiences of our HCWs at highest risk, we can improve our psychological support using targeted interventions during future waves of this pandemic or similar devastating events.

## Introduction

The emergence of a severe acute respiratory coronavirus 2 (SARS-CoV-2) in China at the end of 2019 has led to a global pandemic [1], causing drastic disruptions to social, economic, and

**Competing interests:** The authors have declared that no competing interests exist.

healthcare structures globally. Increasing cases, insufficient amount of personal protection equipment (PPE), extremely demanding workloads, and lack of adequate therapies to save lives can contribute to a psychological burden directly related to working during disease outbreaks [2, 3].

Previous studies on HCWs working during past epidemics have similarly shown an increased mental burden among HCWs [4, 5], and social isolation was a crucial stressor identified. During a sustained outbreak of vancomycin-resistant enterococci (VRE), the staff sustained severe stress due to feeling inadequately supported, feeling blamed for the outbreak, and that they had an increased workload as they took on duties of other staff [6]. A recent meta-analysis investigating the psychological impact of COVID-19 on HCWs demonstrates a high prevalence of anxiety, depression, stress and insomnia explained by uncertainty around the future of the pandemic, availability of a vaccine, increased workload, lack of social support, and fear of familial transmission [7].

In the 2003 SARS outbreak, several stressors for HCWs were identified, including: worrying about infecting family members, feelings of uncertainty, inadequate staff and supplies, personal danger and inability to fight the disease appropriately [8]. A later epidemic caused by another member of the coronavirus family, MERS-CoV, cited similar findings with stressors for HCWs centered around personal safety, well-being of family members and colleague, and watching patients die [9].

Stamford Hospital is a 305 bed Level II Trauma center located in Fairfield County, the county at the epicenter of the pandemic in Connecticut. At the peak of the first wave in mid-April of 2020, the number of infected patients was rising daily requiring the addition of three intensive care units, an additional intermediate care unit and two floors doubling capacity from 36 to 72 beds to isolate confirmed COVID-19 positive patients. During this time, the hospital immediately established a command center to organize teamwork, resources and provide important infectious disease updates. A Joint Military Task Force consisting of the Connecticut National Guard, United States Army and Reserve Corps, worked alongside the physicians, nursing staff, and residents to help meet the increased need to provide high quality critical care brought on by the pandemic.

HCWs are at a high risk of contracting COVID-19 due to its ability to attack human cells and spread so efficiently through asymptomatic and symptomatic individuals. This descriptive study is aimed at exploring emotions and identifying stress and coping strategies of HCWs during the first wave of the pandemic in our hospital. In understanding the needs and experiences of our healthcare providers at highest risk, we hope to provide better psychological support in future waves of this pandemic and future epidemics or pandemics of similar destructive nature.

## Materials and methods

The authors utilized a cross-sectional survey design engaging HCWs in our hospital who worked in high-risk areas from March of 2020 to July of 2020. The study tool is a comprehensive questionnaire derived and modified from the SARS epidemic in 2003 previously described in the literature [8] and the MERS-CoV epidemic in 2014 [9]. A 37-item questionnaire including 5-point Likert scales was utilized with the following responses: Strongly Disagree, Disagree, Neither Agree or Disagree, Agree or Strongly Agree (See S1 File). Questions inquired about demographic information, staff feelings during the COVID-19 pandemic, factors that caused stress among staff during the COVID-19 pandemic, factors that helped reduce stress, personal coping strategies used during the pandemic and motivational factors for future epidemics/ pandemics.

The study questionnaire was distributed electronically and anonymously in August of 2020 to frontline HCWs at high risk for exposure and subsequently contracting COVID-19. The survey comprised of 37 items representing a comprehensive list of respondent emotions, stressors, stress mitigators, and coping mechanisms during the COVID-19 pandemic. Convenience sampling was used to identify our target population from the following departments: Nursing, Respiratory Therapy, Medicine, Surgery, Environmental Services, Transport, Dietary and Emergency Department at Stamford Hospital. Those excluded were employees of Stamford Hospital who were working remotely at any time during the pandemic and employees who were not at risk for significant exposure. Participants were allotted four weeks to complete the survey. De-identified data was collected into a secure database, and subsequently coded. Due to small sample sizes in departments with less employees, Role in Healthcare was combined into three categories: Nurse, Physician and Other.

All analyses were performed using SPSS version 25.0, and prior to study initiation, the study protocol was reviewed and approved as exempt by the Stamford Hospital's Institutional Review Board (IRB) of record. Data analysis was conducted in two phases, reporting of demographic variables as well as descriptive statistics regarding experience and role-based variables. These statistics were reported as count and percent within category. The second phase of this analysis was an analysis of variance (ANOVA), for the three Role in Healthcare groups, (Nurses, Physicians, Other). All survey responses were collected using a Likert type scoring system based on 5 points. A selection of '1' indicated the respondent strongly disagreed with the statement, a selection of '5' indicated the respondent strongly agreed with the statement. Since all categories for all selection items were populated with at least one response, a mean value is a valid reporting indicator for each question. ANOVA results for each item were restricted to only direct patient care providers which included: 131 nurses (60.6%), 21 physicians (9.7%) and 64 (29.6%) respondents working in other related healthcare professions. Statistical significance was found for 15 of the 37 items, however due to the large sample size, small differences in mean values may not have indicated a clinically significant result.

To assess for statistically significant differences between the groups, ANOVA was conducted for each survey item. Reporting of results consisted of the mean and standard deviation within each of the groups, and an omnibus p-value less than 0.05 (p $<$0.05) for the overall ANOVA was considered statistically significant. For the purposes of this report, a greater than 0.50 response difference between highest and lowest mean value will be presented. In the event of a statistically significant omnibus p-value, the Scheffe test was used as the post hoc procedure. Results from all demographic, role-based items and ANOVAs were placed into tables. Not all respondents answered every question, the analysis only included valid responses for each item. There was no missing value imputation conducted for this data. In addition, due to the exploratory nature of this analysis there were no corrections applied to the p-values obtained from the ANOVA statistic due to multiple comparisons. In the case of a significant omnibus test, post hoc results were reported for each of the three combinations of groups and presented accordingly.

## Results

A total of 315 out of 1,976 eligible employees completed the survey, yielding a response rate of 16%. The most frequent response for age category was those aged between 30–39 (27.2%), followed by those in the 40–49 age group (25.6%) (Table 1). The respondents were mostly female (80.8%), and Caucasian (59.0%). The majority of the sample are married (58.0%), with the highest degree attained as a bachelor's degree (24.4%), followed by a BSN(20.3%). Table 1 also presents the results of the working experience and role-based descriptive variables. The survey

**Table 1. Study sample demographic, experience, and role-based characteristics.**

| Variable | Category | Count (n) | Percent (%) |
|---|---|---|---|
| Age | 20–29 | 44 | 17.9% |
| | 30–39 | 67 | 27.2% |
| | 40–49 | 63 | 25.6% |
| | 50–59 | 47 | 19.1% |
| | >60 | 25 | 10.2% |
| Gender | Male | 41 | 16.7% |
| | Female | 202 | 80.8% |
| | Choose not to disclose | 7 | 2.8% |
| Ethnicity | African American | 24 | 9.8% |
| | Caucasian | 144 | 59.0% |
| | Latino/Hispanic | 23 | 9.4% |
| | Asian | 26 | 10.7% |
| | American Indian or Alaska Native | 1 | 0.4% |
| | Two or more races | 2 | 0.8% |
| | Choose not to disclose | 24 | 9.8% |
| Marital Status | Married | 148 | 58.0% |
| | Single | 72 | 28.2% |
| | Divorced/Separated | 21 | 8.2% |
| | Choose not to disclose | 14 | 5.6% |
| Highest Education Level | GED | 6 | 2.4% |
| | Bachelor's Degree | 60 | 24.4% |
| | Diploma in Nursing | 5 | 2.0% |
| | Associates in Nursing | 13 | 5.3% |
| | BSN | 50 | 20.3% |
| | MSN | 41 | 16.7% |
| | DNP | 2 | 0.8% |
| | MD | 22 | 8.9% |
| | Other | 47 | 19.1% |
| Role in Healthcare | Nurse | 143 | 58.1% |
| | Physician | 22 | 8.9% |
| | Other | 81 | 32.9% |
| Years in Healthcare role | 0–5 | 58 | 22.7% |
| | 6–10 | 43 | 16.8% |
| | 11–15 | 37 | 14.5% |
| | 16–20 | 36 | 14.1% |
| | >20 | 82 | 32.0% |
| Years at Stamford | Less than 6 months | 9 | 3.6% |
| | 6 months to 1 year | 23 | 9.1% |
| | 1–3 years | 53 | 21.0% |
| | 3–5 years | 31 | 12.3% |
| | 5–10 years | 33 | 13.1% |
| | >10 years | 103 | 40.9% |
| I provided direct patient care to patients during the COVID-19 pandemic (i.e., my role required me to enter COVID units, patient rooms and/or exposed me to COVID-19 patients). | Yes | 232 | 73.7% |
| | No | 32 | 10.2% |
| | Not Answered | 51 | 16.5% |

participants consisted mostly of registered nurses (58.1%). Thirty-two percent of respondents had more than 20 years' experience in a healthcare role, followed by 0 to 5 years (22.7%). Among participants, 40.9% had worked for more than 10 years at the current hospital and the overwhelming majority (73.7%) reported that they provided direct patient care to COVID-19 patients and had been in units or patient rooms exposed to COVID-19.

## Staff emotions

Those within the nursing profession were more likely to strongly agree with the statement "I felt nervous and scared" than members from physician and other roles (p = 0.002 nursing from physicians, p = 0.04 other roles). Nurses also reported they thought of calling in sick (ANOVA p = 0.007) however all three groups disagreed with that statement. Only 12.1% of staff divulged that they had called in sick at least once during this period, with no differences between groups when analyzed by chi square tests (see S1 File). Additionally, all groups disagreed or strongly disagreed that they would quit their job if another outbreak occurred (Table 2).

## Major stress factors

Nurses were more likely to agree with the statement "I was stressed because I was emotionally exhausted" and "I was stressed because I was physically stressed/fatigued", as compared to their physician cohort (p = 0.001, p = 0.002 respectively). Nurses were also significantly more likely to agree with the statement "I was stressed because I felt there were not adequate protective measures." And "I was stressed because there was a shortage of staff at times" (post hoc test versus physicians: p = 0.008, and p = 0.001 respectively). All groups experienced high levels of stress around potential of transmission of COVID-19 to family and friends and not knowing when the pandemic would be under control (Table 3).

## Effective measures to reduce stress

Those in the nursing profession were more likely to agree with the statement "Getting daily COVID-19 updates from the hospital leadership helped reduce my stress", (post hoc p-value between nursing and physician roles p = 0.004). Nurses were more likely to disagree with the

**Table 2. Staff feelings during the COVID-19 outbreak who were directly involved in taking care of patients.**

| Question | Nursing (1) | | Physician (2) | | Other (3) | | p-value | Post-hoc* |
|---|---|---|---|---|---|---|---|---|
| | Mean | SD | Mean | SD | Mean | SD | | |
| I felt I had to do my job as it is my professional and ethical duty. | 4.54 | 0.84 | 4.95 | 0.21 | 4.67 | 0.59 | 0.033 | 1 vs 2 = 0.045 |
| | | | | | | | | 1 vs 3 = 0.417 |
| | | | | | | | | 2 vs 3 = 0.274 |
| I felt nervous and scared. | 4.30 | 0.92 | 3.45 | 1.34 | 3.81 | 1.09 | <0.001 | 1 vs 2 = 0.002 |
| | | | | | | | | 1 vs 3 = 0.004 |
| | | | | | | | | 2 vs 3 = 0.349 |
| I appreciated the special recognition for my job by hospital administration. | 3.45 | 1.19 | 3.14 | 1.39 | 3.30 | 1.19 | 0.435 | N/A |
| I thought of quitting my job. | 2.24 | 1.33 | 1.77 | 1.15 | 2.04 | 1.29 | 0.233 | N/A |
| I would quit my job if a COVID-19 outbreak recurred. | 1.89 | 1.02 | 1.45 | 0.671 | 1.78 | 0.98 | 0.188 | N/A |
| I thought of calling in sick. | 2.14 | 1.32 | 1.27 | 0.55 | 1.95 | 1.14 | 0.007 | 1 vs 2 = 0.008 |
| | | | | | | | | 1 vs 3 = 0.520 |
| | | | | | | | | 2 vs 3 = 0.070 |

*Pairwise p-values reported.

**Table 3. Questions regarding factors that caused stress among staff during the COVID-19 outbreak.**

| Question | Nursing (1) | | Physician (2) | | Other (3) | | p-value | Post-hoc* |
|---|---|---|---|---|---|---|---|---|
| | Mean | SD | Mean | SD | Mean | SD | | |
| It stressed me to see my colleagues getting sick. | 4.05 | 1.05 | 4.25 | 0.97 | 4.09 | 0.89 | 0.686 | N/A |
| It stressed me to think that I could transmit COVID-19 to my family and friends. | 4.69 | 0.64 | 4.41 | 1.01 | 4.53 | 0.75 | 0.101 | N/A |
| It stressed me to see patients with COVID-19 dying in front of me. | 4.43 | 0.80 | 4.25 | 0.97 | 4.32 | 0.83 | 0.531 | N/A |
| It was stressful not knowing when the COVID-19 pandemic will be under control. | 4.64 | 0.67 | 4.36 | 1.05 | 4.46 | 0.77 | 0.086 | N/A |
| I was stressed because I was Emotionally exhausted. | 4.36 | 0.84 | 3.45 | 1.54 | 4.09 | 1.11 | <0.001 | 1 vs 2 = 0.001 |
| | | | | | | | | 1 vs 3 = 0.158 |
| | | | | | | | | 2 vs 3 = 0.035 |
| I was stressed because I was physically stressed / fatigued. | 4.19 | 0.992 | 3.32 | 1.32 | 3.95 | 1.12 | 0.001 | 1 vs 2 = 0.002 |
| | | | | | | | | 1 vs 3 = 0.274 |
| | | | | | | | | 2 vs 3 = 0.051 |
| I was stressed because I experienced conflict between my duty and my own safety. | 3.92 | 1.27 | 3.27 | 1.42 | 3.69 | 1.25 | 0.066 | N/A |
| I was stressed because I felt there were not adequate protective measures. | 4.22 | 1.08 | 3.36 | 1.50 | 3.60 | 1.31 | <0.001 | 1 vs 2 = 0.008 |
| | | | | | | | | 1 vs 3 = 0.001 |
| | | | | | | | | 2 vs 3 = 0.711 |
| I was stressed because there was a shortage of staff at times. | 4.15 | 1.10 | 3.05 | 1.46 | 3.92 | 1.14 | <0.001 | 1 vs 2 = 0.001 |
| | | | | | | | | 1 vs 3 = 0.372 |
| | | | | | | | | 2 vs 3 = 0.008 |

*Pairwise p-values reported.

statement "my stress reduced because of the protective equipment provided to me by the hospital", whereas both physicians and those in other health care roles tended to agree with that statement (p = 0.012, post hoc p = 0.049). All groups similarly felt stress reduction due to a sense of camaraderie in working together as well as sharing jokes or humor with colleagues (Table 4).

## Coping strategies

Nurses were also more likely to agree with the statement "I talked to myself and motivated myself to face the Covid-19 pandemic with a positive attitude as a personal coping strategy.", than were the physician cohort (post hoc p-value = 0.001. All three cohorts were more likely to disagree with the statement: "I got help from family physicians or other doctors/therapists to reduce my stress and get reassurance." However, physicians were significantly more likely to disagree with that statement (p = 0.031, post hoc). Nurses were more likely to agree with the statement "I vented emotions by crying, screaming etc." than were physicians and those working in other roles who tended to disagree with that statement (having a mean value less than 3.0). This difference achieved statistical significance between nursing and physician roles (p = 0.001). Following strict personal protective measures as well as keeping separate clothes for work to minimize disease transmission were common coping strategies among all groups (Table 5).

## Motivation factors

While all three cohorts agreed with the statement "psychiatric help and therapy made available in the workplace to help reduce stress and anxiety could promote my willingness to participate in any future epidemic/pandemic, those in other health care roles and nursing roles were more likely to agree to that statement than were physicians (p = 0.029, p = 0.032 respectively), and no difference was seen between nursing role respondents and other healthcare role respondents (p = 0.961). All groups agreed or strongly agreed that both financial recognition and

**Table 4. Factors that helped in reducing stress during the COVID-19 outbreak.**

| Question | Nursing (1) | | Physician (2) | | Other (3) | | p-value | Post-hoc* |
|---|---|---|---|---|---|---|---|---|
| | Mean | SD | Mean | SD | Mean | SD | | |
| My stress reduced when I saw improvement in patient's condition. | 4.05 | 1.05 | 4.25 | 0.97 | 4.09 | 0.89 | 0.686 | N/A |
| My stress reduced because of the protective equipment provided to me by the hospital. | 4.69 | 0.64 | 4.41 | 1.01 | 4.53 | 0.75 | 0.101 | N/A |
| My stress reduced because all healthcare professionals were working together on the front line. | 4.43 | 0.80 | 4.25 | 0.97 | 4.32 | 0.83 | 0.531 | N/A |
| My stress reduced because of my confidence in the hospital staff in case I got sick from COVID-19. | 4.64 | 0.67 | 4.36 | 1.05 | 4.46 | 0.77 | 0.086 | N/A |
| My stress reduced when I shared jokes or humor with colleagues | 4.36 | 0.84 | 3.45 | 1.54 | 4.09 | 1.11 | <0.001 | 1 vs 2 = 0.001 |
| | | | | | | | | 1 vs 3 = 0.158 |
| | | | | | | | | 2 vs 3 = 0.035 |
| My stress reduced when I got free meals from the hospital/community | 4.19 | 0.992 | 3.32 | 1.32 | 3.95 | 1.12 | 0.001 | 1 vs 2 = 0.002 |
| | | | | | | | | 1 vs 3 = 0.274 |
| | | | | | | | | 2 vs 3 = 0.051 |
| Getting daily COVID updates from the hospital leadership helped reduce my stress. | 3.92 | 1.27 | 3.27 | 1.42 | 3.69 | 1.25 | 0.066 | N/A |
| Meeting with members of the Army to talk about the stress I was experiencing helped to reduce my stress. | 4.22 | 1.08 | 3.36 | 1.50 | 3.60 | 1.31 | <0.001 | 1 vs 2 = 0.008 |
| | | | | | | | | 1 vs 3 = 0.001 |
| | | | | | | | | 2 vs 3 = 0.711 |

*Pairwise p-values reported.

recognition from management would encourage willingness to participate in future epidemics/pandemics (Table 6).

## Discussion

COVID-19 is an unknown and dangerous virus that the world has not previously faced. To the best of our knowledge, our study is among the first of its kind to explore the emotions,

**Table 5. Personal coping strategies used by the staff to alleviate stress.**

| Question | Nursing (1) | | Physician (2) | | Other (3) | | p-value | Post-hoc* |
|---|---|---|---|---|---|---|---|---|
| | Mean | SD | Mean | SD | Mean | SD | | |
| I followed strict personal protective measures (e.g., mask, face shield, gown, hand washing etc. as a personal coping strategy. | 4.55 | 0.59 | 4.23 | 1.15 | 4.46 | 0.74 | 0.131 | N/A |
| I kept separate clothes for work to minimize transmission as a personal coping strategy. | 4.51 | 0.76 | 4.27 | 1.12 | 4.37 | 0.80 | 0.287 | N/A |
| I did relaxation activities, e.g., involved in prayers, exercise etc., as a personal coping strategy. | 3.90 | 1.03 | 3.67 | 1.49 | 3.94 | 0.98 | 0.575 | N/A |
| I chatted with family and friends to relieve stress and obtain support as a personal coping strategy. | 3.92 | 1.16 | 3.77 | 1.38 | 4.20 | 0.80 | 0.110 | N/A |
| I talked to myself and motivated myself to face the COVID-19 pandemic with positive attitude as a personal coping strategy. | 4.02 | 0.94 | 3.14 | 1.32 | 3.84 | 0.99 | 0.001 | 1 vs 2 = 0.001 |
| | | | | | | | | 1 vs 3 = 0.419 |
| | | | | | | | | 2 vs 3 = 0.015 |
| I got help from family physicians or other doctors/therapists to reduce my stress and get reassurance. | 2.83 | 1.30 | 2.09 | 1.19 | 2.65 | 1.08 | 0.028 | 1 vs 2 = 0.031 |
| | | | | | | | | 1 vs 3 = 0.565 |
| | | | | | | | | 2 vs 3 = 0.165 |
| I avoided media news about COVID-19 and related fatalities as a coping strategy. | 3.16 | 1.29 | 2.68 | 1.56 | 3.25 | 1.20 | 0.188 | N/A |
| I vented emotions by crying, screaming etc. | 3.24 | 1.29 | 2.09 | 1.38 | 2.81 | 1.38 | <0.001 | 1 vs 2 = 0.001 |
| | | | | | | | | 1 vs 3 = 0.074 |
| | | | | | | | | 2 vs 3 = 0.077 |

*Pairwise p-values reported.

**Table 6. Motivation factors promoting willingness to participate in future events of similar nature.**

| Question | Nursing (1) | | Physician (2) | | Other (3) | | p-value | Post-hoc* |
|---|---|---|---|---|---|---|---|---|
| | Mean | SD | Mean | SD | Mean | SD | | |
| Adequate personal protective supplies provided by the hospital could promote my willingness to participate in any future epidemic/pandemics. | 4.31 | 0.99 | 4.64 | 0.58 | 4.32 | 0.88 | 0.304 | N/A |
| Available cure or vaccine for the disease could promote my willingness to participate in any future epidemic/pandemics. | 4.05 | 1.05 | 4.55 | 0.60 | 3.89 | 1.06 | 0.029 | 1 vs 2 = 0.108 |
| | | | | | | | | 1 vs 3 = 0.530 |
| | | | | | | | | 2 vs 3 = 0.029 |
| Financial recognition of efforts could promote my willingness to participate in any future epidemic/pandemics. | 4.39 | 0.94 | 4.09 | 1.11 | 4.53 | 0.81 | 0.137 | N/A |
| Recognition from management and supervisors for the extra efforts could promote my willingness to participate in any future epidemic/pandemics. | 4.10 | 1.02 | 4.00 | 1.16 | 4.35 | 0.82 | 0.141 | N/A |
| Psychiatric help and therapy made available in work place to help reduce stress and anxiety could promote my willingness to participate in any future epidemic/pandemics. | 3.66 | 1.09 | 3.00 | 1.23 | 3.70 | 1.04 | 0.022 | 1 vs 2 = 0.032 |
| | | | | | | | | 1 vs 3 = 0.961 |
| | | | | | | | | 2 vs 3 = 0.029 |
| Reduced working hours during outbreaks could promote my willingness to participate in any future epidemic/pandemics. | 3.64 | 1.08 | 3.27 | 1.35 | 3.59 | 1.08 | 0.358 | N/A |

*Pairwise p-values reported.

perceived stressors, and coping strategies of the HCWs who faced and continue to face the COVID-19 pandemic. Employees all expressed a high level of concern for their ability to transmit COVID-19 to their family members, as well as their personal safety. Humor in the workplace, transparent and frequent communication, availability of PPE, recognition and monetary compensation are all additional important factors to HCWs while combating a sustained pandemic. Of note, our study included HCWs from multiple disciplines who were not represented in previous studies [8, 9].

While there were various stressors related to the first wave of the COVID-19 pandemic studied at our institution, it was most stressful for HCW's to think they could transmit the disease to their family and friends. Uncertainty on knowing when the pandemic would come to an end was identified as a significant stress for all groups as well as seeing patients dying from COVID-19. Importantly, these stressors were also identified in previous studies [8, 9]. The nursing staff was most likely to report feelings of stress from emotional exhaustion and fatigue. This may be due to the differences in direct COVID-19 patient care experiences between nurses and other HCWs. Compared to physicians and other HCWs, nurses spend more time and energy caring for these critically ill patients while also managing their own family members, protection, and general uncertainty. This notion is supported by a recent meta analyses on HCWs during the current pandemic and previous literature on the SARS epidemic [7, 10–12]. The authors found higher levels of anxiety and depression among nurses compared to physicians, concluding that this may be due to the close proximity and exposure potential to a very infectious novel illness [7].

In the current study, most HCWs indicated they did not receive help from family physicians or other doctors/therapists to reduce stress and provide reassurance. Most neither disagreed or agreed that they took part in relaxation activities or exercise to help alleviate stress. Therefore, hospitals should specifically focus on interventions to promote self-care and potentially reduce shift length as suggested in previous outbreaks [5].

When asked about factors which helped reduce stress, all groups were aided by a feeling of camaraderie amongst healthcare professionals working together as well as sharing jokes or

humor with colleagues. This is in line with previous research concluding that positivity and optimism are important at preventing burn out, decreasing emotional exhaustion and improving efficiency [13]. Although this is true, to reduce potential unnecessary exposure to the virus, the hospital discouraged staff from interacting with each other both inside and outside the hospital, and staff meetings were generally all held virtually. This is a time when individuals may wish to seek support from each other but cannot do so, thus potentially increasing the burnout and psychological burdens our HCWs carry.

Nurses did not experience stress relief due to the availability of PPE which corresponded to nurses feeling more stressed than other groups because of the lack of PPE. Even while all groups strongly agreed with following personal protective methods, access to appropriate PPE is paramount in moving forward to reduce HCW stress levels. In fact, availability of adequate PPE is a driving factor for all groups surveyed in their willingness to participate in future epidemics/pandemics.

HCWs are at high risk for COVID-19 exposure, infection and potential illness, however, other studies have shown multiple exposure vectors, both internal and external to the hospital environment, pose a threat. Many HCWs reported symptoms of COVID-19 and 20% had serological indication of infection during the first surge in a recent study published on critical care staff [14]. Using temporal analysis, the authors found that critical care staff were unlikely directly infected by their patients due to the significant availability of PPE worn in these high-risk COVID-19 units. Multidisciplinary staff (therapists, diagnostics, housekeeping, and general groups) working in various locations had the highest seroprevalence, suggesting staff became infected in non-COVID-19 designated areas within the hospital, or from the surrounding community. Regulations of PPE within their institution differed by exposure potential or was not used at all, and participants were two times as likely to be seropositive if they lived with someone who was symptomatic. Due to most staff remaining seronegative even after the first surge, the authors concluded that PPE was effective in protecting staff from COVID-19 [14].

Our results show that the staff experienced emotional trauma during the first wave of the COVID-19 pandemic occurring in Connecticut in early Spring of 2020. The willingness to participate in a second wave of the pandemic or future outbreaks is strongly driven by adequate PPE, financial recognition of efforts and recognition from management as reported in other disease outbreaks [8, 15, 16]. While each outbreak differs in geographic location, transmissibility, and infection and fatality rates, we found that a driving emotion was the professional and ethical duty to perform their job. These findings are similar to those from past epidemics [8, 9].

It is unknown if the COVID-19 pandemic will wane over the next year with vaccine availability and public awareness. We also face uncertainty in knowing if a future epidemic or pandemic will occur. Therefore, it is of utmost importance for healthcare institutions to prepare for the possibility of another epidemic/pandemic of similar nature. Investigation is warranted on the associations between HCW stress and the clinical environment as these associations are cited to span from factors at the individual level (e.g., differences in personality, comorbid conditions) to environmental level variables (e.g., case load, PPE availability) [17].

Indeed, with the recent HCW suicides reported [18–20], hospital incident command centers can include psychiatric preparedness and stress monitoring for health care teams in their planning to understand and address the complex relationships between these variables with clinical health and development of targeted interventions. Specifically, female nurses with high risk for exposure have the most to gain from these efforts [21] though all HCWs should be addressed along with occupational and environmental level factors (importantly, 80% of the current study sample identified as female). Future research directed at determining the effectiveness of interventions to support the mental health of HCWs is a high priority [22]. As the

on-going COVID-19 pandemic provides an opportunity for an evaluation of interventions aimed at supporting our HCWs in these high-risk settings, there is a great need for robust clinical trials developed with appropriate and swift reporting of data to allow for standardized evidence-based implementation of these interventions [23].

Our study is not without limitations. While all HCWs identified as high risk for exposure were invited to participate, the sample size of some respondent groups was not large enough to identify by specific profession or distinguish their responses based on place of work, and 80% of our sample was comprised of females. This is potentially due to some groups with limited access to email, such as Dietary and Environmental Services. Henceforth, our overall participation rate was low; for all who were invited, our response was less than 20%. In addition, the study was conducted three months after the 2020 Spring surge at our hospital, which could result in recall bias. Furthermore, employees who left the organization or who had returned to their normal roles and were missed by the survey distribution lists could result in selection bias. This study presents a single-center experience of the 2020 Spring COVID-19 surge in a Magnet-and Planetree designated community-based Hospital; a smaller or larger non-accredited hospital could have encountered differing scenarios than our institution.

## Conclusion

The COVID-19 pandemic has caused tremendous strain on the healthcare system and its workers who find themselves on the front lines, fighting to treat and contain this virulent disease. Our HCWs faced extreme stress and experienced significant conflict between their duties as HCWs versus safety concerns for themselves as well as their patients, colleagues, and families. The results of this study are similar to those findings reported by staff during the 2003 SARS outbreak, and when facing the MERS-CoV epidemic. Therefore, we can conclude that psychological reactions to extreme stress are common among HCWs caring for patients during highly infectious epidemics/pandemics. The emotions of HCWs working in high-risk environments, their stressors, and how they coped featured distinct elements. By understanding the needs and experiences of our healthcare providers at highest risk, we hope to provide enhanced and targeted psychological support in future waves of this pandemic and during future events of similar destructive nature.

## Supporting information

**S1 File.**
(DOCX)

**S1 Checklist. STROBE statement—checklist of items that should be included in reports of cross-sectional studies.**
(DOCX)

## Acknowledgments

The authors thank Dr. Thomas Wasser for his statistical expertise and support with this manuscript.

## Author Contributions

**Conceptualization:** Suzanne Rose, Seema Pillai.

**Data curation:** Suzanne Rose.

**Investigation:** Suzanne Rose, Josette Hartnett, Seema Pillai.

**Methodology:** Suzanne Rose, Josette Hartnett, Seema Pillai.

**Project administration:** Suzanne Rose, Seema Pillai.

**Resources:** Suzanne Rose, Josette Hartnett.

**Supervision:** Suzanne Rose, Seema Pillai.

**Writing – original draft:** Suzanne Rose, Josette Hartnett, Seema Pillai.

**Writing – review & editing:** Suzanne Rose, Josette Hartnett, Seema Pillai.

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
