## [Decision Letter · Decision Letter 0]

25 May 2021

PONE-D-21-11310

Healthcare Worker’s Emotions, Perceived Stressors and Coping Mechanisms During the COVID-19 Pandemic

PLOS ONE

Dear Dr. Rose,

Thank you for submitting your manuscript to PLOS ONE. After careful consideration, we feel that it has merit but does not fully meet PLOS ONE’s publication criteria as it currently stands. Therefore, we invite you to submit a revised version of the manuscript that addresses the points raised during the review process.

We look forward to receiving your revised manuscript.

Kind regards,

Ismaeel Yunusa, PharmD, PhD

Academic Editor

PLOS ONE

Additional Editor Comments (if provided):

Please submit a revised version along with s STROBE (https://www.equator-network.org/reporting-guidelines/strobe/) checklist for reporting observational cross-sectional studies. Also concise your Introduction to 3-4 paragraphs. This will encourage busy physicians to read your interesting study.

Journal Requirements:

2)  We note that you have indicated that data from this study are available upon request. PLOS only allows data to be available upon request if there are legal or ethical restrictions on sharing data publicly. For more information on unacceptable data access restrictions, please see http://journals.plos.org/plosone/s/data-availability#loc-unacceptable-data-access-restrictions.

3) We note that you have included the phrase “data not shown” in your manuscript. Unfortunately, this does not meet our data sharing requirements. PLOS does not permit references to inaccessible data. We require that authors provide all relevant data within the paper, Supporting Information files, or in an acceptable, public repository. Please add a citation to support this phrase or upload the data that corresponds with these findings to a stable repository (such as Figshare or Dryad) and provide and URLs, DOIs, or accession numbers that may be used to access these data. Or, if the data are not a core part of the research being presented in your study, we ask that you remove the phrase that refers to these data.

Reviewers' comments:

Reviewer's Responses to Questions

**Comments to the Author**

1. Is the manuscript technically sound, and do the data support the conclusions?

Reviewer #1: Yes

Reviewer #2: Partly

2. Has the statistical analysis been performed appropriately and rigorously? 

Reviewer #1: No

Reviewer #2: No

3. Have the authors made all data underlying the findings in their manuscript fully available?

Reviewer #1: Yes

Reviewer #2: No

4. Is the manuscript presented in an intelligible fashion and written in standard English?

Reviewer #1: Yes

Reviewer #2: Yes

5. Review Comments to the Author

Reviewer #1: This is a very timely manuscript exploring the perceived stressors and coping mechanisms as experienced by HCWs, who were in contact with COVID19 patients during the first wave of the COVID-19 pandemic at Stamford hospital.

It is a very detailed questionnaire based cross sectional survey. Although I agree with the statistical methods employed, it is quite confusing to interpret the questions answered by the study participants.

It would be more clear and easy to follow if the questions and its responses were answered as sliding scale with percentages of responses for all the 5 likert scale responses to get a sense of how each question was answered across the whole group or within subgroups specified by the authors.

Reviewer #2: This is an interesting and relevant study. The authors aimed to address a timely research and clinical gap of knowledge among Healthcare workers (HCWs). The paper is generally well written and structured. However, in my opinion the paper has a major limitation in regards to the data analyses. The data were not tested for statistical assumptions for ANOVA such as normality test, etc., prior to the analysis. So, it is not possible to know if the ANOVA used in the analysis was appropriate or not. Using the ANOVA test inappropriately can invalidate the results and the study's conclusion. I will suggest the authors to run the analysis again (to test the assumptions) to see if the ANOVA or Kruskall Wallis is most suitable for this data.

2. The response is very low, and the authors did not mention how the rate was calculated in the methods section. the authors failed to weigh the study samples, the literature suggests "weighing" a study sample when a response rate is around 20% (Jack 2008).

(https://www.ncbi.nlm.nih.gov/pmc/articles/PMC2384218/#:~:text=Response%20rates%20approximating%2060%25%20for,of%20%E2%89%A5%2080%25%20is%20expected.)

The authors should state the efforts made to improve the response rate.

3. In the methods section, it was not clear how the data was collected? was it face-to-face, via email etc.

4.No sample size calculation.

5. Over 80% of the sample were females. This should be mentioned in the limitation section.

6. I will suggest the first paragraph of the discussion section to explain the key findings and how it answer the research questions.

7. The conclusion sectin should be brief and focus on the main study findings.

6. PLOS authors have the option to publish the peer review history of their article (what does this mean?). If published, this will include your full peer review and any attached files.

Reviewer #1: No

Reviewer #2: **Yes: **Ibrahim Jatau Abubakar

---

## [Author Response · Author response to Decision Letter 0]

11 Jun 2021

Please see the document entitled Response to Reviewers for our responses to the editor and reviewers' comments. Thank you

---

## [Decision Letter · Decision Letter 1]

24 Jun 2021

Healthcare Worker’s Emotions, Perceived Stressors and Coping Mechanisms During the COVID-19 Pandemic

PONE-D-21-11310R1

Dear Dr. Rose,

We’re pleased to inform you that your manuscript has been judged scientifically suitable for publication and will be formally accepted for publication once it meets all outstanding technical requirements.

Kind regards,

Ismaeel Yunusa, PharmD, PhD

Academic Editor

PLOS ONE

Additional Editor Comments (optional):

In tables 2-6, kindly have table footnotes where you clarify that in the last column (Post-hoc), what you are reporting is pairwise p values. Also, replace 'v' with 'vs'. Tables should be self-explanatory, which may include clarifications in a footnote whenever necessary.

Reviewers' comments:

Reviewer's Responses to Questions

**Comments to the Author**

1. If the authors have adequately addressed your comments raised in a previous round of review and you feel that this manuscript is now acceptable for publication, you may indicate that here to bypass the “Comments to the Author” section, enter your conflict of interest statement in the “Confidential to Editor” section, and submit your "Accept" recommendation.

Reviewer #2: All comments have been addressed

2. Is the manuscript technically sound, and do the data support the conclusions?

Reviewer #2: (No Response)

3. Has the statistical analysis been performed appropriately and rigorously? 

Reviewer #2: (No Response)

4. Have the authors made all data underlying the findings in their manuscript fully available?

Reviewer #2: (No Response)

5. Is the manuscript presented in an intelligible fashion and written in standard English?

Reviewer #2: (No Response)

6. Review Comments to the Author

Reviewer #2: (No Response)

7. PLOS authors have the option to publish the peer review history of their article (what does this mean?). If published, this will include your full peer review and any attached files.

Reviewer #2: **Yes: **Ibrahim Jatau Abubakar

---

## [Editor Report · Acceptance letter]

28 Jun 2021

PONE-D-21-11310R1 

Healthcare Worker’s Emotions, Perceived Stressors and Coping Mechanisms During the COVID-19 Pandemic 

Dear Dr. Rose:

I'm pleased to inform you that your manuscript has been deemed suitable for publication in PLOS ONE. Congratulations! Your manuscript is now with our production department. 

Kind regards, 

on behalf of

Dr. Ismaeel Yunusa 

Academic Editor

PLOS ONE